# Research on the Impact of Rural Land Transfer on Non-Farm Employment of Farm Households: Evidence from Hubei Province, China

**DOI:** 10.3390/ijerph192315587

**Published:** 2022-11-24

**Authors:** Lili Chen, Jiquan Peng, Yibei Zhang

**Affiliations:** School of Economics, Jiangxi University of Finance and Economics, Nanchang 330013, China

**Keywords:** rural land transfer, non-farm employment, propensity score matching method (PSM), generalized propensity score matching method (GPSM)

## Abstract

Agricultural scale operations and industrialization promote the transfer of the rural labor force to the industry sector, and the non-farm employment of farmers plays a great role in increasing their income and reducing poverty. It is of great significance to explore the non-farm employment of farmers for the governance of relative poverty and the achievement of common prosperity. The propensity score matching (PSM) and generalized propensity score matching (GPSM) were used to analyze the impact of rural land transfer on farm households’ non-farm employment. According to the PSM estimation, compared to the farmers’ land not transferred, the rural land transfer significantly increased the proportion of non-farm employment personnel in farm households and the months of per year non-farm employment per person. The total land transfer, paddy land transfer and dry land transfer could significantly increase the proportion of non-farm employment personnel in farm households by 0.074, 0.029 and 0.085 units, respectively, and could significantly increase the months of per year non-farm employment per person by 0.604, 0.394 and 0.617 units, respectively. According to the GPSM estimation, different types of rural land transfer areas have significant positive effects on the proportion of non-farm workers and the months of per year non-farm employment per person, and show an obvious increasing trend of returns to scale, that is, the proportion of non-farm workers and the months of per year non-farm employment per person of farmers are higher than the increase in rural land transfer area. Additionally, the return to scale effect of dry land transfer area is more obvious. In order to raise the income of farm households and narrow the gap between urban and rural areas, the land transfer system can be further improved, urbanization with the county town as an important carrier can be vigorously promoted, the participation of farm households in non-farm employment in the local area can be promoted and the support policy system for non-farm employment of rural labor force can be improved.

## 1. Introduction

More than 40 years of reform and opening up in China has greatly promoted the process of new industrialization and urbanization. A large number of jobs created in cities and towns have laid the conditions for the transfer of rural labor force, and the surplus rural labor force released by the progress of agricultural technology has provided the possibility for its transfer. As a result, more and more rural labor force is transferred to cities to engage in non-farm work, and the livelihood of farmers is gradually transformed from pure agricultural operation to part-time operation mode, which strongly promotes the increase in farmers’ income and the elimination of absolute poverty. With the completion of the comprehensive prosperous society, the realization of “common prosperity” has become a vivid practice and action guide of the great rejuvenation of the Chinese nation, but according to the *China Statistical Yearbook 2021*, the lowest per capita disposable income of 20% of the Chinese residents in 2020 was only USD 1111.8, and the per capita disposable income of 40% of the people was around USD 141.3 per month, indicating that the vast majority of the relatively poor are still rural residents. Additionally, as of 2020, more than 36% of China’s population lives in rural areas, but the agricultural output value accounted for only is less than 7%, which means that nearly 64% of the urban population produces and distributes 93% of the GDP, while the other 36% produces and distributes less than 7% of the GDP, which indicates that the gap between urban and rural areas is the biggest obstacle to China’s realization of “common prosperity”. The fundamental solution to the problems of “agriculture, rural areas and farmers” in China has never been found in rural areas but in urban areas. Land and labor are the only livelihood capital of farmers, and how to use the livelihood capital to continuously improve the sustainability of their livelihoods is the key. In the case of the current low agricultural comparative advantage, the method of raising farmers’ income in the short term is still expanding farmers’ employment. With the continuous improvement of the level of agricultural modernization, the technical effect of capital replacing labor will gradually increase, and the increasingly perfect rural land transfer market will further release the rural labor attached to the land and continue to move to the cities along with China’s urbanization. Therefore, it is of great practical significance to explore the impact of rural land transfer on farmers’ non-farm employment for improving the sustainability of farmers’ livelihoods and achieving common prosperity.

## 2. Literature Review and Theoretical Mechanisms

### 2.1. Literature Review

Land and labor are important factors of production for farmers, a relationship that many scholars have thoroughly discussed, but more scholars focus on the analysis of non-farm employment as an influence of land transfer [1,2,3], and only a few scholars analyze the impact of land transfer on non-agricultural employment of rural households [4,5], mainly from the perspective of the evolution of the land system and the development of the transfer market [6]. The non-farm employment of rural labor force is closely related to the rural land system [7], as the adjustment of contracted land management rights will reduce farmers’ expectation of the stability of land rights [8,9], make farmers divest their possession of land use rights and turn to invest more labor force in the field of non-farm production [10,11,12]. Subsequently, many scholars found that land right confirmation can increase the expectation of pure farming farmers on the stability of land property rights and eliminate their concerns after land transfer [13,14,15]; moreover, the land scale effect will improve agricultural production efficiency [16], thus promoting the employment of rural labor force [17]. However, some scholars have found that land right confirmation can enhance farmers’ right to dispose of land, improve the comparative advantage of agricultural production by reconfiguring agricultural production factors and improve farmers’ expectation of future agricultural income and enthusiasm for agricultural production [18], thus increasing the investment in land, indirectly increasing the opportunity cost of non-agricultural employment and restraining the willingness of land transfer and non-agricultural employment. Furthermore, considering the social security function of land, farmers will not easily transfer their land to engage in non-farm employment [19,20]. Studies have shown that the land confirmation will accelerate the development of the land transfer market [21], as the mature land market development and stable market lease will increase the liquidity of the land use right [22,23], promote the non-farm employment of the rural labor force [24], but also, scholars believe, the active land rental market will increase the degree of farmland scale and operating earnings [25], farmers may not give up agricultural production to engage in non-farm employment and high land transfer costs will also inhibit farmers from engaging in non-farm employment [26]. 

At present, a large number of studies have essentially confirmed that non-farm employment will promote land transfer [27,28], that land fragmentation will strengthen the role of non-farm labor supply on farmland outflow and that participation in the new rural cooperative medical endowment insurance or commercial endowment insurance will also play a promoting role [29,30]. However, some scholars believe that the inter-generational division of the labor model of half-work and half-farming and the inter-generational relay citizenship model will have an impact [31]. In recent years, some scholars analyzed the impact of land transfer to non-farm employment, and their conclusions essentially show that land transfer can promote agricultural machinery engaged in non-farm employment [32,33] and also, influenced by the scale of operation and the degree of mechanization [34], the labor transfer effects of the transfer of cultivated land, woodland, pastureland and the housing land present evident differences [35]. In summary, at present, more studies focus on the impact of non-farm employment on land transfer [36,37], but there are a few studies directly analyzing the impact of land transfer on non-farm employment, as previous studies have not paid enough attention to their endogeneity, leading to questionable research results. In addition, previous studies only concluded that land transfer would promote the farmers’ participation in non-farm employment, but did not carefully analyze the impact of land transfer on the number and time of non-farm employment of farmers. 

Generally speaking, the allocation of family labor force includes the quantity of labor and labor time. Based on the field survey data, this paper uses the GPSM model to analyze the impact of land transfer on the proportion of non-agricultural employees and per capita non-agricultural employment time. Considering the different effects of different types of land transfer, this paper further divides the types of land transfer into paddy field and dry land, thus exploring the effects of different types of land transfer of non-agricultural employment.

### 2.2. Mechanisms of the Impact of Land Transfer on Non-Farm Employment of Farm Households

Land transfer can promote large-scale agricultural operations and enable farmers to transfer family labor to non-farm industries with greater confidence on the basis of retaining land contract rights. The specific mechanisms are as follows: First, land transfer can reduce the economic threshold of rural labor transfer [38,39,40]. As a rational subject, smallholders will consider their expected benefits and economic costs when choosing labor transfer [41], the expected benefits of rural labor transfer being the sum of rental income from land transfer and non-farm income [42,43], and the economic cost of rural labor transfer being the sum of the opportunity cost of abandoning agricultural production and the cost of transferring to non-farm industries [44]. The transfer costs of rural labor to non-farm industries include the intermediary costs of land transfer, travel expenses for returns home to farm in busy season and labor fees, etc. [45] The premise of the transfer of rural labor is that the expected income exceeds the economic cost at a certain point, which is the “economic threshold” [46]. It can be seen that a perfect land transfer market and intermediary organization can help farmers reduce the cost of land transfer, so as to reduce the “economic threshold” of labor transfer [47]. Second, the land transfer can reduce the social threshold of rural labor transfer [48,49]. “Social threshold” refers to the rural labor force adapting to the needs of urban life, ideas and life skills and other necessary conditions. Although the reform and opening-up as well as basic compulsory education have greatly improved the cultural literacy of rural residents [50,51], as farmers have been living in a relatively closed rural society for a long time, their cognition and awareness may not be immediately integrated into the urban lifestyle. [52,53,54]. The land transfer system helps farmers break the self-sufficiency mode of production and management [55], participate in social production and division of labor on a larger scale, accept new urban ideas and concepts and lower the “social threshold” for rural labor to transfer to urban employment. 

Third, land transfer can improve agricultural labor productivity to promote the rural labor transfer [56]. The traditional decentralized management mode will have many adverse effects on farmers’ production [57], such as financing difficulties, farmers’ agricultural machinery use not being convenient, less advanced agricultural science and technology application, etc. [58,59,60], which hinder the improvement of the efficiency of agricultural production. Ranis-Fei has also suggested that low labor productivity severely reduces the likelihood of labor migration. Land transfer can concentrate land resources to those who are willing to farm or are capable of farming [61,62], improve the utilization rate of agricultural machinery through continuous scale operation, better apply new technology in the production process, improve agricultural labor productivity and release the surplus labor force in the countryside [63,64,65]. Fourth, land transfer promotes the rural labor transfer by weakening the land safeguard function [66,67]. Land is the most important means of production for farmers, and farmers can only obtain the materials they need to live by engaging in agricultural production [68,69]. Therefore, the land plays the role of social security for rural residents in China [70,71], which is of great significance to the alleviation of social contradictions and conflicts. Therefore, farmers generally do not easily hand over their land to others. With the rise of land transfer, farmers can not only obtain land rent and non-farm labor income after land transfer, but also keep the contractual right of land, as land rent and non-farm income are far higher than agricultural income [72], which weakens the social security function of land and makes farmers more willing to transfer land [73]. 

## 3. Research Methods and Data Processing

### 3.1. Model Construction

The influence of land transfer on the non-farm employment of farmers can be estimated by building an econometric model, but the problems of sample self-selection and endogeneity that may exist in the estimation model need to be solved [74]. In addition, the increase in land transfer area will release more rural labor originally attached to the land, so as to promote the non-farm employment degree of farmers; on the contrary, when the degree of non-farm employment of farmers increases, it will also be difficult for farmers to engage in agricultural production due to insufficient labor force, so they will transfer their land to others for planting. Therefore, from a theoretical perspective, land transfer and the degree of non-farm employment may be mutually causal. Moreover, the difference in the initial endowment between households with and without land transfer may lead to the problem of sample self-selection in the estimation of the model. At present, the random effect model, fixed effect model, DID model and instrumental variable method are usually used to solve the sample self-selection and endogeneity problems that may exist in econometric models, but the first three models are not suitable for cross-sectional data, and it is difficult to obtain appropriate instrumental variables. 

Propensity score matching (PSM) can better estimate the treatment effect of binary variables and overcome the endogeneity problem caused by sample self-selection to a certain extent. PSM is based on the principle of “Counterfactual framework”, supposing that the land-transferring households are the treatment group, and the land-non-transferring households are the control group, matching the control group and the treatment group according to the propensity score value. The sample characteristics of the control group and the treatment group were as similar as possible; then, the control group was used to simulate the non-transfer of the treatment group, and finally, the difference of the non-agricultural employment degree of the farmers after the land transfer were compared. This paper uses the proportion of non-farm workers and the months of per year non-farm employment per person to measure the degree of non-farm employment. Based on the generalized propensity score matching model used by Peng and Chen [74], this paper establishes an econometric model. The operation idea is as follows: the binary selection model is used to estimate the probability of land transfer for each sample under the condition of given eigenvector, which is the propensity score (PS) value, and its expression is as follows:(1)$${\mathrm{PS}}_{i}=\mathrm{Pr}\left[D_{i}=1{|x}_{i}\right]=E\left[D_{i}=0{|x}_{i}\right]$$

In Equation (1), $D_{i}=1$ indicates that farm households participate in land transfer, and $D_{i}=0$ indicates that farm households do not participate in land transfer. $x_{i}$ is the household characteristics that can be observed. In the counterfactual case, the difference in the degree of non-farm employment of farmers under the transferred and not transferred land was compared, and the difference was the net treatment effect. The average effect (ATT) of farmers’ non-farm employment degree is:(2)$$\mathrm{ATT}=E\left[y_{1i}-y_{0i}{|D}_{i}=1\right]=\frac{1}{N_{t}}\sum_{i\in t}y_{t}^{i}-\frac{1}{N_{t}}\sum_{j\in t}\lambda \left(p_{i},p_{j}\right)y_{c}^{j}$$

In Equation (2), refers to the land transfer household, T is the matched experimental group, c becomes the control group before matching, $y_{c}^{i}$ is the observed value of the i-th land transfer household in the treatment group, $y_{c}^{j}$ is the observed value of the j-th household whose land has not transferred in the control group, $p_{i}$ is the predicted probability value of farmer i in the experimental group, $p_{j}$ is the predicted probability value of farmer j in the control group, $\lambda \left(p_{i},p_{j}\right)$ is the weight function; different matching methods correspond to different weight functions. 

PSM can only estimate the difference of non-agricultural employment degree between land-transferred households and land-non-transferred households, but it cannot obtain the causal relationship between transferred area and non-agricultural employment degree of rural households; therefore, there needs to be an evaluation model that can handle key variables as continuous variables. In order to solve the above problems, Hirano and Imbens extended PSM to generalized propensity score matching (GPSM) [75]. GPSM can estimate the impact of land transfer area on the non-farm employment of farmers at any treatment level. The operation ideas are as follows:

First, the maximum-likelihood method is used to estimate the conditional probability distribution $G\left(T_{i}\right)$ of the continuous treatment variable T when the covariate X is given:(3)$$G\left(T_{i}\right)|X_{i}\sim N\left(y\left(\lambda X_{i}\right),\sigma^{2}\right)$$

In Equation (3), $y\left(\lambda X_{i}\right)$ is the linear function of covariate X; λ and $\sigma^{2}$ are the parameters to be estimated. The generalized propensity score is estimated from the covariate X as follows:(4)$${\;\hat{P}}_{i}=\frac{1}{\sqrt{2\pi \;{\hat{\sigma }}^{2}}}\exp\left\{-\frac{1}{2{\;\hat{\sigma }}^{2}}\left[G\left(T_{i}\right)-y\left(\lambda X_{i}\right)\right]\right\}$$

Second, the treatment variable T and the generalized propensity score ${\;\hat{P}}_{i}$, which is estimated by Equation (4), are used to construct the model; then, the conditional expectation (i.e., the degree of non-farm employment of farmers) of the result variable $F_{i}$ is obtained:(5)$$E\left({\;\hat{P}}_{i}{|T}_{i},{\;\hat{P}}_{i}\right)=\gamma_{0}+\gamma_{1}T_{i}+\gamma_{2}T_{i}^{2}+\gamma_{3}{\;\hat{P}}_{i}+\gamma_{4}{\;\hat{P}}_{i}^{2}+\gamma_{5}T_{i}{\;\hat{P}}_{i}$$

In Equation (5), the role of ${\;\hat{P}}_{i}$, ${\;\hat{P}}_{i}^{2}$ and $T_{i}{\;\hat{P}}_{i}$ is to eliminate the problems of endogeneity and sample selection bias in the model. 

Finally, the regression result of Equation (4) is substituted into Equation (5); then, the expected value of the result variable $F_{i}$ when the processing variable is T can be obtained as follows:(6)$$\;\hat{E}\left[F\left(t\right)\right]=\frac{1}{N}\sum_{i=1}^{N}\left[{\;\hat{\gamma }}_{0}+{\;\hat{\gamma }}_{1}t+{\;\hat{\gamma }}_{2}t_{i}^{2}+{\;\hat{\gamma }}_{3}\;\hat{p}\left(t,X_{i}\right)+{\;\hat{\gamma }}_{4}\;\hat{p}{\left(t,X_{i}\right)}^{2}+{\;\hat{\gamma }}_{5}t\;\hat{p}\left(t,X_{i}\right)\right]$$

In Equation (6), N is the sample observation value and $\;\hat{p}\left(t,X_{i}\right)$ is the conditional probability density prediction value of the treatment variable. The range of values of the treatment variable $\;\bar{T}=\left[t_{0},t_{1}\right]$ is divided into n subintervals $\;\bar{T}=\left(n=1,2,\cdots ,n\right)$, at each of which the causal effect of land transfer area on the degree of non-farm employment of farmers can be estimated separately. By connecting the causal effects under each range of values, a graph of the functional relationship between the size of the causal effect and the land transfer area in the interval $\;\bar{T}\left(t_{0},t_{1}\right)$ can be derived. 

### 3.2. Data Sources

The data in this paper are from the research group’s field survey of farmers in Hubei Province in 2018, which covers the basic information of household population, natural and material assets, production and management, land transfer behavior and farmers’ policy cognition, etc. The survey sites are in Jianli County and Qichun County, and the survey area types include plains, hills and mountainous areas, indicating that the selected survey sites can ensure the representativeness of the samples. Moreover, Qichun County has 4193.36 hectares of arable land, with a total population of 1.034 million. Jianli County has 116,000 hectares of arable land and a total population of 1.546 million. The proportion of farmland transfer in the two counties exceeded the average level of farmland transfer in Hubei Province, and a large amount of rural labor force went out to work. The use of these two counties as analysis samples is more conducive to exploring the impact of land transfer on non-agricultural employment of rural labor. The survey data involved 11 towns and 44 villages; 26 farmers were surveyed in each village, a total of 1144 households were investigated, 24 invalid questionnaires were eliminated and a total of 1120 valid samples were obtained. The author analyzed the sample data and found that 24 sample households had no contracted farmland, as the farmland of these households had been expropriated by the government in the process of urbanization. Therefore, these 24 samples do not meet the research requirements of this paper and were deleted. In addition, the data in this study do not violate ethical principles.

### 3.3. Variable Selection

Explained variable: The explained variable in this paper is non-farm employment. Generally speaking, rural labor force refers to the group aged 16–65 years with labor abilities (excluding students). When the rural labor force is engaged in production and operation activities other than agriculture, it means that the rural labor force is in the state of non-farm employment. According to neoclassical economics, the career choice of labor force is based on the principle of maximizing its own income. When allocating labor resources, Chinese rural families will make reasonable decisions according to their own specific conditions and those of the other members of their family. Therefore, the measurement of non-farm employment needs to be conducted from the household level in order to further investigate the depth and breadth of rural labor force engaged in non-farm employment. It is planned to consider whether there are non-farm workers in the household, the proportion of non-farm workers in the household and the months of per year non-farm employment per person of the household for analysis. 

Explanatory variables: This paper mainly analyzes the impact of land transfer on non-farm employment, including whether the land is transferred and land transfer area to investigate the two aspects, and land transfer types which are subdivided into paddy land and dry land, finally generating three dummy variables of whether the land is in transfer and three kinds of land transfer area continuous variables. 

Control variables: The control variables selected in this paper include the characteristics of the household head, family characteristics and regional characteristics, as follows: The education level of the household head. The household head is the main decision maker of household production and management, and his or her cultural literacy will affect the rationality of decision. Family social capital. Rural labor force is usually introduced to non-farm production and management activities by acquaintances or relatives in the village, and family social capital generally increases the possibility of migrant work. Family burden coefficient. Overburdened families will force family labor to engage in more profitable non-farm production and management activities. Unhealthy membership ratio. Unhealthy members not only reduce the number of people in the labor force in the family, but also reduce the working hours of healthy workers who care for unhealthy members. Number of agricultural machineries. Agricultural machinery is an important material for agricultural production and operation. The more agricultural machinery there is, the more likely are farmers to engage in agricultural production and operation. Enthusiasm for farming. Farmers with high enthusiasm for farming are less likely to seek non-farm employment. Region type. Land resources and production conditions of farmers in different regions are not consistent, which affects their management decisions. 

### 3.4. Descriptive Statistical Analysis

Table 1 shows the descriptive statistics of the variables. The average education level of the household head is 6.889 years, but the variance is large, indicating that the number of education years of the household head is not very high, and the difference is large. The average value of family social capital is 7.368, and the large variance indicates that the expenditure of human relations between farmers is quite different. The average household burden coefficient is 0.313, indicating that almost one-third of the people in each household require support. The mean proportion of unhealthy members was 0.166, with a small variance, indicating that unhealthy people were prevalent in farm households, but the difference in the number was small. The mean value of the quantity of agricultural machinery is 1.007, and a large variance indicates that farmers have a large difference in agricultural fixed capital. The mean value of farming enthusiasm was 0.483, indicating that most farmers were not sufficiently active in farming. The mean value of district type was 0.333, indicating that one-third of the sample was located in a plain area. 

## 4. Results and Analysis

### 4.1. The PSM Estimation of Non-Farm Employment of Farm Households in the Case or Absence of Land Transfer

#### 4.1.1. Sample Matching Effect Test

Before using the propensity score matching method, it is necessary to test the balance of the matching variables between the treatment group and the control group. The usual practice is to verify the balance test results and the effect map of propensity value distribution. Two kinds of test results are listed in this paper; Table 2 shows the balance test results of propensity score matching, and Figure 1 shows the kernel density distribution map of propensity score before and after matching. Labor allocation includes labor quantity and labor time. Land transfer will also have an impact on the amount and time of non-farm employment of farmers. The following is the matching test result of whether land transfer affects the proportion of non-farm employment personnel in farm households and the months of per year non-farm employment per person. Table 2 shows that regardless of whether the dependent variable in the model is the proportion of non-farm employment personnel in farm households or the months of per year of non-farm employment per person, in the control variables, there are significant differences in household head education level, family social capital, family burden coefficient, proportion of unhealthy members, number of agricultural machineries, agricultural enthusiasm and regional type, the mean difference of each covariate before matching being higher than that after matching. The result of propensity score matching is more scientific and reliable only when there is no significant difference in matching variables between land transfer households and land non-transfer households after matching. The method to determine the matching effect is to compare whether the standard deviation of each covariate before and after matching is of less than 20%. Table 2 shows that the standard deviation of each matching variable is below 20%, indicating that the matching effect is ideal. This study also tested the balance of matching results between paddy land transfer and dry land transfer, and the test results show that both of them met the balance assumption, which was not listed in this paper due to space limitation. 

Then, using the method of graphic, visual inspection matching effect, as seen in in Figure 1, (a) and (b) are the test charts of the proportion of non-farm employment personnel in farm households as the dependent variable, (c) and (d) are test charts wherein the dependent variable is the months of per year non-farm employment per person, and the kernel density before and after matching the of the PS values of households with land transfer (treatment group) and households without land transfer (control group) are given. From graphs (a) and (b), the probability distributions of propensity scores for the matched pre-treatment group and the control group differed significantly, regardless of whether the dependent variable in the model was the proportion of non-farm workers or the duration of non-farm employment per capita; after matching, the two distributions are essentially the same. The results show that the characteristics of the post-processing group and the control group are similar, and the matching results satisfy the common supporting hypothesis required by the propensity score matching method. According to the type of land transfer, this paper also compares the effect maps of propensity value distribution before and after matching between dry land transfer households and paddy land transfer households. The results show that the matching results, which are not listed in the article, are good.

#### 4.1.2. The PSM Estimation of the Proportion of Non-Farm Employment Personnel in Farm Households in the Case or Absence of Land Transfer

The propensity score matching method has a variety of estimation methods; therefore, to estimate the robustness of the results, this article also uses the radius matching method, the kernel density matching method and the Mahalanobis matching method to estimate the average treatment effect of whether the land is transferred, whether the paddy land is transferred and whether the dry land is transferred on the proportion of farmers’ non-farmer employment separately. The specific estimation results are shown in Table 3. From the average treatment effect of whether land is transferred on the proportion of non-farm employment personnel in farm households, land transfer before matching can significantly increase the proportion of non-farm employment personnel in farm households by 0.071 units, and after matching, the estimated ATT values of the three matching methods are 0.074, 0.073 and 0.075 units, respectively, and the estimated results are all significant at the 1% level. In general, after eliminating sample differences by the matching method, the average coefficient of ATT value of net effect of land transfer is 0.074, which is significantly larger than the estimated coefficient before matching, indicating that sample estimation bias will underestimate the effect of land transfer. Compared with that without land transfer, land transfer can increase the proportion of non-farm employment personnel in farm households. This shows that land transfer can release the rural labor force attached to agricultural production to a greater extent, so that rural families will allocate more labor force in the field of non-agricultural production and management.

From the average treatment effect of whether paddy field is transferred on the proportion of non-farm employed persons in farm households, a paddy field transferred before matching will significantly increase the proportion of non-farm employed persons in farmers by 0.028 units, the estimated ATT values of the three matching methods after matching are 0.029, 0.029 and 0.030 units, respectively, and the estimated results are all significant at the 1% level. The average coefficient of ATT value of paddy field net effect was 0.029. From the average treatment effect of whether dry land is transferred on the proportion of non-farm employment personnel in farm households, dry land transfer before matching can significantly increase the proportion of non-farm employment personnel in farm households by 0.082 units, the estimated ATT values of the three matching methods after matching are 0.086, 0.084 and 0.084 units, respectively, and the estimated results are all significant at the 1% level. The average coefficient of ATT value of net effect of dry land transfer was 0.085. The average effect of dry land transfer was more than twice that of paddy transfer, showing that dry land transfer can increase the proportion of non-farm employment personnel in farm households more than paddy land transfer. This is mainly due to the fact that drylands grow a greater variety of crops and that dryland crops are more cumbersome and time-consuming than paddy-field crops in the agricultural production process. Therefore, dry land transfer can release more rural labor force. 

#### 4.1.3. The PSM Estimation to the Months of per Year of Non-Farm Employment per Person in the Case or Absence of Land Transfer

To continue with the estimation method described above, the average treatment effects of whether the land is transferred, whether the paddy land is transferred and whether the dry land is transferred on the non-farm employment per capita time of farmers are estimated, and the specific estimation results are shown in Table 3. From the average treatment effect of potential land transfer on the months of per year of non-farm employment per person of farmers, land transfer before matching can significantly increase the months of per year non-farm employment per person of farmers by 0.503 units. After matching, the estimated ATT values of the three matching methods are 0.607, 0.606 and 0.600 units, respectively, and the estimated results are all significant at the 1% level. In general, after eliminating sample differences by the matching method, the average coefficient of ATT value of net effect of land transfer is 0.604, which is significantly larger than the estimated coefficient before matching, indicating that sample estimation bias will underestimate the effect of land transfer. Compared to the land not being transferred, land transfer can improve the months of per year non-farm employment per person of farmers, indicating that land transfer can greatly reduce the agricultural production time of rural families, so that rural families can allocate more labor time in the field of non-farm production and management. 

From the average treatment effect of potential paddy land transfer on the months of per year non-farm employment per person of farmers, the transfer of paddy land before matching will significantly increase the months of per year non-farm employment per person of farmers by 0.375 units. The estimated ATT values of the three matching methods after matching are 0.390, 0.392 and 0.400 units, respectively, and the estimated results are all significant at the 1% level. The average coefficient of ATT value of paddy land net effect was 0.394. According to the average treatment effect of potential dry land transfer on the months of per year non-farm employment per person of farmers, dry land transfer before matching can significantly increase the months of per year non-farm employment per person of farmers by 0.537 units. The estimated ATT values of the three matching methods after matching are 0.612, 0.616 and 0.624 units, respectively, and the estimated results are all significant at the 1% level. The average coefficient of ATT value of net effect of dry land transfer was 0.617. It can be seen that the average effect of dry land transfer is more than double that of paddy land transfer, indicating that dry land transfer can improve the months of per year non-farm employment per person of farmers more than paddy land transfer. This may be due to the fact that drylands are more complex and time-consuming to manage than paddy fields, the drylands thus releasing more rural labor.

### 4.2. The GPSM Estimation of Land Transfer Area on Non-Farm Employment of Farm Households

It has been verified above that land transfer can improve the proportion of farmers’ non-farm employment more than that without land transfer, but the impact of land transfer area on the proportion of farmers’ non-farm employment is still unclear. Therefore, this paper uses GPSM to analyze the causal relationship between land transfer area and non-agricultural employment.

#### 4.2.1. Balance Tests for Matching Estimates of Generalized Propensity Scores

Before using the GPSM analysis, we first test whether the treatment variable T needs to meet the assumption of normal distribution. By testing the skewness and kurtosis of the distribution of land transfer area, paddy land transfer area and dry land transfer area, it is found that the three treatment variables obey the null hypothesis of normal distribution. Then, the fractional logit model was used to estimate the generalized propensity score and test whether the covariates adjusted by the generalized propensity score passed the balance test. According to the balance test idea of Hirano and Imbens (2004), the samples need to be grouped. Since the treatment variable land transfer area takes values in [0, 1] after polarization, the division thresholds chosen are 0.1 and 0.5 of the treatment intensity according to the principle of subdividing the intervals of smaller treatment intensity and coarse division of the intervals of larger treatment intensity. According to the partition critical values, in this paper, three types of land transfer samples are divided into three groups, by comparing the mean of one covariate in any group with the mean of the other two groups combined to judge the balance of the covariate; the absence of a significant difference indicates that the equilibrium test is passed. Table 4 summarizes the estimated values of covariates in each group where the dependent variable is the proportion of non-farm workers. The estimation results show that all covariates are not significant after the adjustment of generalized propensity score, which indicates that the matching effect is good and passes the balance test. 

Then, according to the above ideas, the balance of the matching of the generalized propensity score with the dependent variable being the time of non-farm employment per capita is tested. Table 5 summarizes the estimated values of each group of covariates in various land transfer samples. By observing the distribution of treatment intensity, 0.1 and 0.5 are still selected as the dividing criteria. The estimation results show that there is no significant difference between the mean of a covariate in any group and the mean of the other two combinations. The estimation results show that all covariates are not significant after the generalized propensity score adjustment, indicating that the matching effect is good and that the balance test is passed. 

#### 4.2.2. GPSM Estimation of Land Transfer Area on the Proportion of Non-Farm Employment Personnel in Farm Households

Table 6 shows the GPSM estimation results of land transfer area on the proportion of non-farm employed persons. According to the formula, the expected value and marginal change of the proportion of non-farm employed farmers in different types of land transfer area at different treatment levels can be estimated. The results of the second-order approximation show that the total land transfer area and its square, the propensity score variable and its square as well as their interaction terms all pass the significance test. The transfer area of paddy land and its square, propensity score variable and its square as well as their interaction terms all passed the significance test. The dry land transfer area and its square, propensity score variable and its square as well as their interaction terms all passed the significance test, indicating that the second-order approximation estimation method was more suitable. As can be seen from Table 6, the treatment effects of all land transfer area, paddy land transfer area and dry land transfer area are all positive, and there is no linear relationship between the transfer area of each type of land and the proportion of non-farm employment personnel in farm households. The treatment effects of the three types of land transfer area show an increasing trend with the increase in transfer area. However, the marginal effect of different land transfer areas is different in different treatment levels. From the perspective of the total land transfer area, the proportion of farmers’ non-farm employment increases slightly with the increase in land transfer area at the treatment levels of 0–0.7; when the land transfer area exceeds the treatment level of 0.7, the increase in land transfer area increases the proportion of farmers’ non-farm employment more visibly. 

From the perspective of paddy land and dry land transfer area, the effect of land transfer area on the proportion of non-farm employment personnel in farm households was not obvious within the treatment levels of 0–0.7. After the 0.7 treatment level, the increase in land transfer area can more significantly increase the proportion of non-farm employment personnel in farm households. In general, the increase in land transfer area has an increasing return to scale effect on the proportion of farm households with non-farm employment. That is, as the area of land transfer increases, the proportion of non-farm employed persons in farm households increases more than the area of land transfer. The possible explanation is that the transfer area has a threshold effect on the rural household labor force engaged in non-agricultural production and operation activities. That is, when the area of farmland transferred out by farmers is relatively small, rural families still need a large amount of labor force to farm land, and it is unlikely that there will be surplus labor force engaged in non-agricultural production and operation activities. Only when the area transferred out by farmers is large enough, the time spent by rural families on farmland will be greatly reduced, and more surplus labor will be engaged in non-agricultural production and operation activities.

Moreover, at each treatment level, the improvement effect of the proportion of non-farm employment personnel in farm households brought by the increase in dry land transfer area is greater than that of paddy land transfer, indicating that dry land transfer can better promote the increase in the proportion of non-farm employment personnel in farm households. The possible reasons are as follows: paddy lands are generally planted with field crops, and the use of agricultural machinery is high, and while the economies of scale and mechanized operation can effectively save labor, dry land generally has more varieties of crops, and the degree of mechanization is low, leading to the operation of dry land requiring more labor. 

Through the expected value and marginal effect estimated by the propensity score matching method, the treatment effect function of different types of land transfer area on the proportion of non-farm employment personnel in farm households can be obtained. In order to ensure the validity of the estimation results, the treatment effect function of each GPSM was estimated by the Bootstrap method after 500 repetitions. In order to improve the simplicity of this paper, the estimated figures are not shown in the article. In general, all types of land transfer area have a positive impact on the proportion of non-farm employed persons in farm households, and the proportion of non-farm employed persons is increasing with the increase in land transfer area, but there is an evident increasing benefit of return to scale. 

#### 4.2.3. GPSM Estimation of Land Transfer Area on the Months of per Year Non-Farm Employment per Person

Table 7 shows the GPSM estimation results of land transfer area on the months of per year non-farm employment per person. It is found that the second-order approximation estimation method is more suitable. The estimation results show that the treatment effect of all land transfer areas, paddy field transfer areas and dry land transfer areas, is positive, and that the relationship between land transfer areas of all types and the months of per year non-farm employment per person is nonlinear. The treatment effects of the three are increasing with the increase in the transfer area, but the marginal effects of various land transfer areas at different treatment levels are different. From the perspective of the total land transfer area, the effect of land transfer on the months of per year non-farm employment per person differed significantly before and after the treatment level of 0.7. The increase in land transfer area at the treatment levels 0–0.7 has little effect on the improvement of the months of per year non-farm employment per person. When the land transfer area exceeds 0.7, the increase in land transfer area has a greater effect on the improvement of the months of per year non-farm employment per person. 

In terms of the transfer area of paddy land and dry land, the treatment effect was still significantly different before and after the treatment level of 0.7. Within the treatment levels from 0 to 0.7, the transfer area of paddy land and dry land had little effect on the improvement of the months of per year non-farm employment per person. After the treatment level of 0.7, the increase in paddy land and dry land transfer area can increase the months of per year non-farm employment per person more. In general, the increase in paddy field and dry land circulation area has an increasing trend of scale returns on the months of per year non-farm employment per person. The possible explanation for this phenomenon is that the land transfer area has a certain threshold effect on the rural household labor force to engage in non-agricultural production and operation activities. That is, when the land transfer area of farmers is too small, it is not evident to reduce the time of family farming, and families still need to spend more labor and time to manage land. When the land transfer area reaches a certain value, farm households have more time to engage in non-farm production and operation activities. In addition, regardless of the treatment level, the increase in dry land transfer has a greater impact on the increase in the months of per year non-farm employment per person than the impact on paddy land transfer, indicating that dry land transfer is more conducive to an increase in the months of per year non-farm employment per person. This further proves that the operation of dry land requires more labor time, and that the transfer of dry land can promote farmers to participate in non-agricultural employment.

The resulting expectations and marginal effects were then estimated by the propensity score matching method, enabling the treatment effect functions for different types of land transfer area on the months of per year non-farm employment per person to be derived. In order to improve the simplicity of this paper, the estimated figures are not shown in the article. In general, the impact of each type of land transfer area on the months of per year non-farm employment per person is positive, and the months of per year non-farm employment per person is increasing with the increase in land transfer area, but there is an evident increasing benefit of return to scale. 

## 5. Conclusions

Common prosperity should not leave farmers behind, as sustainable livelihoods of farm households are a prerequisite for raising farmers’ incomes and narrowing the urban–rural gap, as well as an important means to promote common prosperity. Based on the field survey data of farm households in 2018, this paper uses the PSM and GPSM models to analyze the impact of land transfer on non-farm employment in farm households. The results show that: (1) According to the estimation by the propensity score matching method, land transfer has a significant positive effect on the proportion of farmers on non-farm employment and the months of per year non-farm employment per person, and all land transfer, paddy land transfer and dry land transfer can significantly increase the proportion of farmers’ non-farm employment by 0.074, 0.029 and 0.085 units, respectively. All land transfer, paddy land transfer and dry land transfer can significantly increase the months of per year non-farm employment per person by 0.604, 0.394 and 0.617 units, respectively. (2) It can be estimated by the generalized propensity score matching method that the land circulation area has a significant positive effect on the proportion of farmers’ non-farm employment and the months of per year non-farm employment per person, and that both are increasing as the area of different types of land transfers increases, with a clear trend of increasing returns to scale. That is to say, the increase in the proportion of farm households’ non-farm employment and the months of per year non-farm employment per person are higher than that of land transfer area. 

## 6. Discussion

This paper finds that farmland transfer can indeed promote rural family labor force to participate in non-agricultural production and operation activities, which is essentially consistent with the existing research results [32,33], and also shows the reliability of this study. In addition, previous studies only focused on the impact of farmland circulation on farmers’ participation in non-agricultural production and operation activities. However, we did not pay attention to the impact of the area of farmland transfer [22,23], which was in the end more conducive to our understanding of the relationship between the transfer of farmland and the allocation of rural labor force [34], as well as conducive to the formulation of corresponding policies by government decision makers [35]. This further shows that it is very necessary to select the generalized propensity score matching method for research in this paper, and scholars can continue to use this method when analyzing such problems. With the continuous expansion of land transfer scale, more rural labor force has the opportunity to participate in non-farm production and operation activities such as going out to work, and non-farm income is more conducive to increase farm household income and achieve common prosperity. First, we will further improve the land transfer system by appropriately extending the contract period on the basis of the current land contract period, so that farmers can participate in the land transfer with greater confidence. We will rely on rural management institutions to improve the land transfer service platform, enable the flow of information about supply and demand on both sides of transfer and guide the orderly transfer of land management rights. Second, we will vigorously develop the county economy and provide more local non-farm employment opportunities for rural labor. According to the “Opinions on Promoting Urbanization Construction with County Seat as an Important Carrier”, local governments take the county as the basic unit to promote the integrated development of urban and rural areas, which can not only effectively solve the employment problem of rural migrant labor, but also avoid a series of problems caused by the failure of family functions caused by labor going out for work. Third, it is necessary to formulate relevant support policy systems for rural labor force, take various forms to strengthen the training of non-farm labor force and improve professional skills and post competence. We need to improve the basic pension insurance system for urban and rural residents and the basic rural medical insurance system that integrates urban and rural areas, constantly raise the level of pensions and medical insurance and help to ease the worries of farmers who have transferred their land. At the same time, take multiple initiatives to build an integrated urban–rural educational protection system for children, and ensure that the children of non-farm workers have equal access to compulsory education. 

## Figures and Tables

**Figure 1 ijerph-19-15587-f001:**
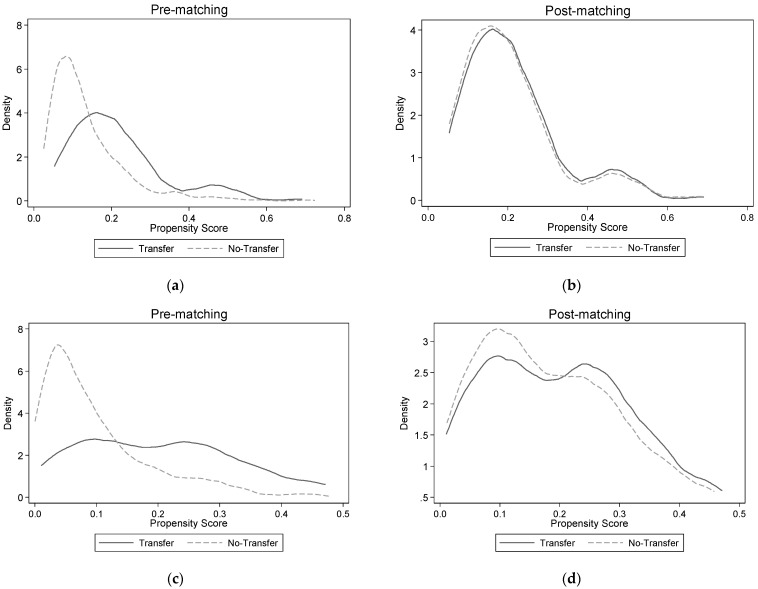
Kernel density distribution of propensity scores before (**a**,**c**) and after (**b**,**d**) matching.

**Table 1 ijerph-19-15587-t001:** Variable definitions and descriptive statistics.

Variable	Computation Method	Mean	Variance	Min	Max
Whether the household is non-farm employed	1 = non-farm employed persons in the household; 0 = no non-farm employed persons in the household	0.896	0.306	0	1
Proportion of non-farm employment personnel in farm household	Non-farm payrolls/household size	0.350	0.202	0	1
The months of per year non-farm employment per person	Total household non-farm time/household size (months/person)	4.069	1.821	0	12
Whether the land is transferred	1 = transferred; 0 = not transferred	0.168	0.374	0	1
Whether the paddy land is transferred	1 = transferred; 0 = not transferred	0.136	0.344	0	1
Whether the dry land is transferred	1 = transferred; 0 = not transferred	0.045	0.209	0	1
Land transfer area	Actual value/hm^2^	0.227	2.082	0	1
Paddy land transfer area	Actual value/hm^2^	0.187	2.215	0	1
Dry land transfer area	Actual value/hm^2^	0.041	1.469	0	0.400
Education level of the head of household	Actual value/year	6.889	3.465	0	15
Social capital of the family	Logarithm of expenses for favors/yuan	7.368	3.085	0	12.337
Family burden factor	Number of elderly and children/household size	0.313	0.287	0	1
Proportion of unhealthy members	Number of unhealthy people/household size	0.166	0.315	0	1
Number of agricultural machines	Actual value/unit	1.007	1.316	0	12
Activism in farming	1 = active; 0 = inactive	0.483	0.500	0	1
Type of region	1 = plain; 0 = non-plain	0.333	0.471	0	1

**Table 2 ijerph-19-15587-t002:** Results of the balance test for propensity score matching.

Variables	Category	The Proportion of Non-Farm Employment Personnel in Farm Households				The Months of per Year Non-Farm Employment per Person			
		Mean		Deviation%	Reduction%	Mean		Deviation%	Reduction%
		Treatment Group	Control Group			Treatment Group	Control Group		
Education level of the head of household	U	7.428	6.817	16.7	93.9	6.170	7.000	−20.3	38.7
	M	7.428	7.465	−1		6.170	5.662	12.5	
Social capital of the family	U	9.740	9.320	30.3	81.3	9.410	9.379	2.3	46.1
	M	9.740	9.818	−5.7		9.410	9.393	1.2	
Family burden factor	U	7.837	7.287	18.5	94.1	6.691	7.453	−22.8	96.3
	M	7.837	7.869	−1.1		6.691	6.663	0.8	
Proportion of unhealthy members	U	0.268	0.321	−19.2	64.3	0.399	0.302	30.7	71.2
	M	0.268	0.249	6.9		0.399	0.427	−8.9	
Number of agricultural machines	U	2.758	1.968	56.5	77.2	1.134	2.203	−81	97.1
	M	2.758	3.078	−12.9		1.134	1.165	−2.3	
Activism in farming	U	1.688	0.889	53	68.9	0.402	1.083	−63.1	95.5
	M	1.688	1.289	16.5		0.402	0.433	−2.9	
Type of region	U	0.586	0.466	24.2	67.5	0.371	0.497	−25.6	59.2
	M	0.586	0.547	7.9		0.371	0.320	10.5	

**Table 3 ijerph-19-15587-t003:** PSM estimation of whether land is transferred on non-farm employment.

Category	Method	Proportion of Non-Farm Employment in Farm Household				The Months of per Year Non-Farm Employment per Person			
		Treatment Group	Control Group	ATT Value	Standard Error	Treatment Group	Control Group	ATT Value	Standard Error
Whether the land is transferred	Not matched	0.415	0.345	0.071 ***	0.024	3.180	2.677	0.503 ***	0.091
	Match 1	0.415	0.341	0.074 ***	0.025	3.180	2.573	0.607 ***	0.097
	Match 2	0.415	0.342	0.073 ***	0.025	3.180	2.574	0.606 ***	0.073
	Match 3	0.415	0.34	0.075 ***	0.028	2.631	2.031	0.600 ***	0.081
Whether the paddy land is transferred	Not matched	0.381	0.353	0.028 **	0.013	2.652	2.277	0.375 ***	0.067
	Match 1	0.381	0.352	0.029 **	0.013	2.652	2.262	0.390 ***	0.075
	Match 2	0.381	0.352	0.029 **	0.013	2.598	2.206	0.392 ***	0.071
	Match 3	0.381	0.351	0.030 **	0.015	2.731	2.331	0.400 ***	0.060
Whether the dry land is transferred	Not matched	0.427	0.345	0.082 ***	0.021	8.533	7.996	0.537 ***	0.075
	Match 1	0.427	0.341	0.086 ***	0.024	8.533	7.922	0.612 ***	0.060
	Match 2	0.427	0.343	0.084 ***	0.023	8.496	7.880	0.616 ***	0.048
	Match 3	0.433	0.348	0.084 ***	0.025	8.530	7.906	0.624 ***	0.055

Note: ** and *** represent significance at the 5% and 1% levels.

**Table 4 ijerph-19-15587-t004:** Generalized propensity score matching balance test for the proportion of non-farm employment personnel in farm households.

Variables	Land Transfer Area			Paddy Land Transfer Area			Dry Land Transfer Area		
	[0, 0.1]	(0.1, 0.5]	(0.5, 1]	[0, 0.1]	(0.1, 0.5]	(0.5, 1]	[0, 0.1]	(0.1, 0.5]	(0.5, 1]
Education level of the head of household	−0.097	0.081	0.206	−0.163	−0.009	0.086	0.023	0.174	−0.463
	(0.170)	(0.289)	(0.185)	(0.204)	(0.352)	(0.211)	(0.370)	(0.421)	(0.400)
Social capital of the family	0.040	−0.351	0.452	−0.053	−0.154	0.559	−0.626	0.807	−0.015
	(0.352)	(0.571)	(0.361)	(0.439)	(0.690)	(0.407)	(0.686)	(0.811)	(0.758)
Family burden factor	−0.427	0.372	0.585	−0.243	0.138	0.218	−0.646	1.101	−0.541
	(0.363)	(0.648)	(0.413)	(0.420)	(0.886)	(0.465)	(0.732)	(0.971)	(0.915)
Proportion of unhealthy members	0.054	−0.061	−0.050	0.047	−0.094	−0.022	0.040	−0.202	−0.011
	(0.041)	(0.054)	(0.035)	(0.037)	(0.067)	(0.040)	(0.073)	(0.190)	(0.085)
Number of agricultural machines	0.015	−0.036	−0.032	−0.002	0.043	0.007	0.062	−0.193	0.032
	(0.036)	(0.063)	(0.041)	(0.043)	(0.077)	(0.046)	(0.078)	(0.198)	(0.092)
Activism in farming	−0.367	0.303	0.544	−0.369	0.504	0.491	−0.704	0.900	0.607
	(0.250)	(0.277)	(0.371)	(0.295)	(0.341)	(0.299)	(0.480)	(0.612)	(0.400)
Type of region	−0.002	0.204	−0.080	−0.036	0.254	−0.042	−0.077	0.136	0.023
	(0.060)	(0.203)	(0.065	(0.072)	(0.226)	(0.075)	(0.131)	(0.157)	(0.148)

Note: Figures in brackets are standard errors.

**Table 5 ijerph-19-15587-t005:** Generalized propensity score matching balance test for the months of per year non-farm employment per person.

Variables	Land Transfer Area			Paddy Land Transfer Area			Dry Land Transfer Area		
	[0, 0.1]	(0.1, 0.5]	(0.5, 1]	[0, 0.1]	(0.1, 0.5]	(0.5, 1]	[0, 0.1]	(0.1, 0.5]	(0.5, 1]
Education level of the head of household	−0.219	0.452	0.131	0.030	−0.339	0.222	0.025	0.012	−0.538
	(0.195)	(0.254)	(0.255)	(0.224)	(0.411)	(0.219)	(0.376)	(0.454)	(0.409)
Social capital of the family	−0.185	0.366	0.451	0.356	−0.538	0.497	−0.625	0.806	−0.028
	(0.419)	(0.516)	(0.514)	(0.509)	(0.814)	(0.440)	(0.699)	(0.864)	(0.766)
Family burden factor	−0.326	0.989	0.442	−0.236	−0.726	0.555	−0.643	−0.490	0.214
	(0.409)	(0.554)	(0.563)	(0.450)	(0.905)	(0.465)	(0.703)	(1.003)	(0.918)
Proportion of unhealthy members	0.055	−0.135	0.001	0.021	0.057	−0.032	0.040	−0.212	−0.003
	(0.034)	(0.146)	(0.047)	(0.039)	(0.077)	(0.041)	(0.071)	(0.195)	(0.086)
Number of agricultural machines	0.002	−0.026	0.014	−0.033	0.072	−0.018	0.062	−0.070	−0.001
	(0.041)	(0.056)	(0.056)	(0.045)	(0.089)	(0.048)	(0.075)	(0.102)	(0.092)
Activism in farming	−0.351	0.447	0.413	−0.327	0.444	0.422	−0.703	1.016	0.496
	(0.268)	(0.236)	(0.232)	(0.218)	(0.401)	(0.306)	(0.475)	(0.745)	(0.405)
Type of region	0.034	−0.115	0.016	−0.002	0.160	−0.056	−0.078	0.071	0.109
	(0.067)	(0.090)	(0.090)	(0.078)	(0.147)	(0.077)	(0.132)	(0.169)	(0.150)

**Table 6 ijerph-19-15587-t006:** GSPM estimation of types of land transfer area on the proportion of non-farm employment personnel in farm households.

Treatment Level	Land Transfer Area		Paddy Land Transfer Area		Dry Land Transfer Area	
	Treatment Effects	Standard Error	Treatment Effects	Standard Error	Treatment Effects	Standard Error
0.1	0.272 ***	0.076	0.263 ***	0.062	0.006 ***	0.126
0.2	0.453 ***	0.095	0.404 ***	0.079	0.229 ***	0.178
0.3	0.711 ***	0.089	0.605 ***	0.068	0.585 ***	0.178
0.4	1.048 ***	0.072	0.868 ***	0.057	1.076 ***	0.153
0.5	1.463 ***	0.070	1.191 ***	0.078	1.700 ***	0.138
0.6	1.957 ***	0.094	1.575 ***	0.112	2.458 ***	0.157
0.7	2.528 ***	0.125	2.021 ***	0.134	3.349 ***	0.191
0.8	3.178 ***	0.150	2.527 ***	0.127	4.375 ***	0.214
0.9	3.906 ***	0.169	3.094 ***	0.090	5.535 ***	0.224
1	4.712 ***	0.202	3.722 ***	0.108	6.828 ***	0.274

Note: Figures in brackets are standard errors, *** represent significance at the 1% level.

**Table 7 ijerph-19-15587-t007:** GSPM estimation of types of land transfer area on the months of per year non-farm employment per person.

Treatment Level	Land Transfer Area		Paddy Land Transfer Area		Dry Land Transfer Area	
	Treatment Effects	Standard Error	Treatment Effects	Standard Error	Treatment Effects	Standard Error
0.1	0.006 ***	0.002	0.263 ***	0.020	1.319 ***	0.255
0.2	0.229 ***	0.011	0.404 ***	0.067	2.168 ***	0.354
0.3	0.585 ***	0.055	0.605 ***	0.051	3.186 ***	0.354
0.4	1.076 ***	0.032	0.868 ***	0.271	4.372 ***	0.319
0.5	1.700 ***	0.439	1.191 ***	0.225	5.726 ***	0.314
0.6	2.458 ***	0.072	1.575 ***	0.110	7.248 ***	0.358
0.7	3.349 ***	0.832	2.021 ***	0.425	8.939 ***	0.406
0.8	4.375 ***	0.717	2.527 ***	0.570	10.799 ***	0.425
0.9	5.535 ***	0.727	3.094 ***	0.144	12.826 ***	0.478
1	6.828 ***	0.862	3.722 ***	0.447	15.023 ***	0.745

Note: Figures in brackets are standard errors, *** represent significance at the 1% levels.

## Data Availability

The data presented in this study are available on request from the corresponding author.

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
