# Peer review of "Research on the Impact of Rural Land Transfer on Non-Farm Employment of Farm Households: Evidence from Hubei Province, China"

_ijerph, 2022, doi:10.3390/ijerph192315587_

Round 1
Reviewer 1 Report
This manuscript studies the impact of rural land transfer on farm households' non-farm employment at household scale by using Propensity score matching (PSM) and generalized propensity score matching (GPSM). It finds: 1) the rural land transfer significantly increased the proportion of non-farm employment personnel in farm households and the months of per year non-farm employment per person; 2) the land transfer area for farm household labor engaged in non-farm production and business operation activities has certain threshold effect.
The structure of the paper is more reasonable, the data and methods are presented fairly clear. However, the introduction and literature part, explanation of the results, and discussion need to be improved.
Comments:
1. As in China there are the varied land transfer through crop attributes, commodity chains, agrarian history and state-corporate activities, it may explain a bit more about selected two counties (Jianli and Qichun), why choose them for case analysis? What are their economic and social backgrounds? What is significant of this study for scientific and policy implications?
2. Analysis and result parts need to be shorten: reducing repetition the narratives about PSM and GPSM which have been already discussed in the methodology; reducing tables and figures (this part there are 6 tables and large 3 figures);
3. The conclusions consistent with the evidence and arguments presented. But one point Author should give an attention. In the section 4.2.2 (GPSM estimation of land transfer area on the proportion of non-farm employment personnel in farm households), author states that “the land transfer area for farm household labor engaged in non-farm production and business operation activities has certain threshold effect, namely when the area of land transferred to farmers is too small, it does little to free up the labour attached to the land and farm households still need to spend more labour and time to manage the land; When the land transfer area reaches a certain value, farm households will have more labor force engaged in non-farm production and operation activities”. This point it may make the significance clear, and needs to be indicate how it leads to wider understanding.
Author Response
Response Letter
We would like to thank the editors and the review team for your constructive comments and suggestions on our paper. With a concerted effort to address all of the valuable advice, we have revised the paper to a more accurate and concise style, providing more information for analyzing the impact of rural land transfer on non-farm employment of farm households. We believe that the recommended changes have dramatically enhanced the manuscript’s quality and its contribution, and we hope you will agree.
Also, it should be noted that we highlight the important changes in our revised manuscript using a revision model. Below, we provide our detailed responses in tabular form to explain how your points have been included in the revision. As you will see, we have made every possible attempt to address your concerns, and we hope you will find our revision acceptable. Once again, thank you for helping us to improve the manuscript greatly.
Comments:The structure of the paper is more reasonable, the data and methods are presented fairly clear.
Responses:Thank you very much for your appreciating the manuscript. As you suggested, we have made considerate efforts to improve the expression and generality of the results of our study. The details are shown in the revised manuscript. The author also revised the introduction and literature part, explanation of the results, and discussion. See the revised version for details.
1. As in China there are the varied land transfer through crop attributes, commodity chains, agrarian history and state-corporate activities, it may explain a bit more about selected two counties (Jianli and Qichun), why choose them for case analysis? What are their economic and social backgrounds? What is significant of this study for scientific and policy implications?
Responses:Thanks for your suggestion. The author further explains the reasons for using these two counties as research samples, as follows: Moreover, Qichun County has 4,193.36 hectares of arable land, with a total population of 1.034 million. Jianli County has 116,000 hectares of arable land and a total population of 1.546 million. The proportion of farmland transfer in the two counties exceeded the average level of farmland transfer in Hubei Province, and a large number of rural labor force went out to work. The use of these two counties as analysis samples is more conducive to exploring the impact of land transfer on non-agricultural employment of rural labor.
2. Analysis and result parts need to be shorten: reducing repetition the narratives about PSM and GPSM which have been already discussed in the methodology; reducing tables and figures (this part there are 6 tables and large 3 figures).
Responses:Thank you for these constructive suggestions. The author has reduced the repetition of PSM and GPSM in the article. In order to enhance the brevity of the article, the author reduces Figure 2 and Figure 3, because the estimated results of these two figures have been shown in the table.
3. The conclusions consistent with the evidence and arguments presented. But one point Author should give an attention. In the section 4.2.2 (GPSM estimation of land transfer area on the proportion of non-farm employment personnel in farm households), author states that “the land transfer area for farm household labor engaged in non-farm production and business operation activities has certain threshold effect, namely when the area of land transferred to farmers is too small, it does little to free up the labour attached to the land and farm households still need to spend more labour and time to manage the land; When the land transfer area reaches a certain value, farm households will have more labor force engaged in non-farm production and operation activities”. This point it may make the significance clear, and needs to be indicate how it leads to wider understanding.
Responses:Thank you very much for your suggestion. The author rewrote the passage to make it easier for readers to understand. The details are as follows: The possible explanation is that the transfer area has a threshold effect on the rural household labor force engaged in non-agricultural production and operation activities. That is, when the area of farmland transferred out by farmers is relatively small, rural families still need a large number of labor force to farmland, and it is unlikely that there will be surplus labor force engaged in non-agricultural production and operation activities. Only when the area transferred out by farmers is large enough, the time spent by rural families on farmland will be greatly reduced, and more surplus labor will be engaged in non-agricultural production and operation activities.

Reviewer 2 Report
Thanks to the authors for their paper. Very informative and accurate. Good and informative abstract. Some suggestions for improvements:
1) I would strongly recommend to make some references to the information sources in the Introduction part as some figures and data are presented. Sums in yuan must be shown also in some more international currency (e.g. USD or EUR) as not all potential readers will be able to estimate and understand easily the value of yuan.
2) If the authors of the paper have not developed the statistical models themselves they use in their research, they must refer to some sources, at least refer to some similar studies. If the authors have developed the models themselves, it must be clearly stated.
3) Why there were "24 invalid questionnaires" (pp.6)? This should be explained. Also methodological description MUST comprise information about the research ethics principles in both data collection and data analysis process.
4) pp.7: "Household head is the main decision maker of household production and management, and his cultural literacy will affect the rationality of decision." Does the statement mean that all household heads are men? If not, more inclusive language form should be used in the paper, such as "..., and his/her...."
5) Discussion part (5.2.) is actually written like conclusions. Discussion means that your results are compared with other studies, some references must be added, some implications for the future studies must be made.
6) In general, the author have used appropriate information sources and literature; however, more international experience would raise the quality of the paper.
Author Response
Response Letter
We would like to thank the editors and the review team for your constructive comments and suggestions on our paper. With a concerted effort to address all of the valuable advice, we have revised the paper to a more accurate and concise style, providing more information for analyzing the impact of rural land transfer on non-farm employment of farm households. We believe that the recommended changes have dramatically enhanced the manuscript’s quality and its contribution, and we hope you will agree.
Also, it should be noted that we highlight the important changes in our revised manuscript using a revision model. Below, we provide our detailed responses in tabular form to explain how your points have been included in the revision. As you will see, we have made every possible attempt to address your concerns, and we hope you will find our revision acceptable. Once again, thank you for helping us to improve the manuscript greatly.
Comments:Thanks to the authors for their paper. Very informative and accurate. Good and informative abstract.
Responses:Thank you very much for your appreciating the manuscript. As you suggested, we have made considerate efforts to improve the expression and generality of the results of our study. The details are shown in the revised manuscript.
1. I would strongly recommend to make some references to the information sources in the Introduction part as some figures and data are presented. Sums in yuan must be shown also in some more international currency (e.g. USD or EUR) as not all potential readers will be able to estimate and understand easily the value of yuan.
Responses:Thanks for your suggestion. The author has expressed all the amounts in the introduction in US dollars according to the opinions of the reviewers.
2. If the authors of the paper have not developed the statistical models themselves they use in their research, they must refer to some sources, at least refer to some similar studies. If the authors have developed the models themselves, it must be clearly stated.
Responses:Thank you for these constructive suggestions. The author added references to the econometric model to enhance the readability of the article. See the revised version for details.
3. Why there were "24 invalid questionnaires" (pp.6)? This should be explained. Also methodological description MUST comprise information about the research ethics principles in both data collection and data analysis process.
Responses:Thank you very much for your suggestion. The author explains why there is an invalid questionnaire, as follows: The author analyzed the sample data and found that 24 sample households had no contracted farmland, and the farmland of these households had been expropriated by the government in the process of urbanization. Therefore, these 24 samples do not meet the research requirements of this paper, and will be deleted. In addition, the data in this study do not violate ethical principles.
4. pp.7: "Household head is the main decision maker of household production and management, and his cultural literacy will affect the rationality of decision." Does the statement mean that all household heads are men? If not, more inclusive language form should be used in the paper, such as "..., and his/her...."
Responses:Thank you for these constructive suggestions. According to the opinions of the reviewers, the author has expressed the head of the household in the text as "he or she" to strengthen the inclusiveness and standardization of the language.
5. Discussion part (5.2.) is actually written like conclusions. Discussion means that your results are compared with other studies, some references must be added, some implications for the future studies must be made.
Responses:Thank you very much for your suggestion. The author revised the discussion section, focusing on the differences between my research results and other studies, and made some suggestions for future research. See the revised version for details.
6. In general, the author have used appropriate information sources and literature; however, more international experience would raise the quality of the paper.
Responses:Thanks for your suggestion. According to the opinions of reviewers, the author cited more international experience to improve the quality of the article. See the revised version for details.
